# LLM-Guider: A Language-Guided Discovery of Symbolic Pruning Metrics for Post-Training Sparsity in LLMs

**Jędrzej Hasiura[1], Prashant Bhat[1], Elahe Arani[1,2], Bahram Zonooz[1]**

**[1]Eindhoven University of Technology (TU/e), Netherlands**
**[2]Wayve Technologies Ltd, London, United Kingdom**

**Reviewed on OpenReview:** `https://openreview.net/forum?id=SlVQxEiYnY`

## Abstract

Large Language Models (LLMs) have achieved remarkable advancements in natural language understanding, yet their mammoth size, coupled with substantial training and inference costs, can make them difficult to use in environments with limited resources. To address both memory and efficiency concerns, post-training unstructured sparsity techniques have emerged, focusing on developing optimal pruning criteria to eliminate redundant weights while maintaining performance. However, these approaches often rely on manually crafted pruning criteria, leading to sub-optimal solutions due to heuristic oversimplifications. Therefore, we introduce LLM-Guider, a language-guided symbolic formula optimization framework that seeks to discover optimal pruning criteria through a transparent and systematic process. LLM-Guider comprises three interrelated stages: example selection, formula generation, and formula evaluation, which collectively enable the efficient exploration of the formula space. In addition, LLM-Guider enables the incorporation of intuition, domain, and mathematical knowledge through role prompts, hints, and in-context examples. We also extend the standard set of aggregation strategies over a calibration dataset, resulting in never-seen-before pruning metrics. Through extensive experiments, we demonstrate that formulas discovered through LLM-Guider is able to find formulas that outperform established baselines.

## 1 Introduction

Large Language Models (LLMs) have demonstrated remarkable natural language understanding and generation abilities with unprecedented accuracy and depth. The impressive performance of LLMs can be largely attributed to their scale, which depends on model parameters, dataset size, and amount of compute used for training (Kaplan et al., 2020). The scaling laws have enabled the development of large models such as GPT-175B (Brown et al., 2020) and beyond, boasting hundreds of billions of parameters. Although this has led to the emergence of new abilities in LLMs Wei et al. (2022a), the associated extraordinary training and inference costs pose major challenges for practical use, especially in resource-constrained settings. In order to address both memory and efficiency concerns, multiple model compression methods such as sparsity, quantization, and knowledge distillation, have been proposed in the literature (Zhu et al., 2024). Model sparsity, either structured or unstructured, essentially focuses on pruning redundant weights while balancing performance versus model size trade-off. As model sparsity involves either training from random initialization (Hoang et al., 2023), retraining (Chen et al., 2023a), or extensive iterative pruning (Tanaka et al., 2020), post-training sparsity approaches (Sun et al., 2024; Zhang et al., 2024; Dong et al., 2024; Frantar & Alistarh, 2023) have become increasingly popular.

The essence of post-training unstructured sparsity techniques involves the development of an optimal pruning criterion that quantifies the significance of each weight, subsequently eliminating those weights that exhibit

the lowest significance scores. A vast majority of approaches proposed in the literature require manual crafting of pruning criterion by utilizing weight magnitude (Cheng et al., 2024), activations (Sun et al., 2024; Zhang et al., 2024), and first/second order gradient information (Das et al., 2024; Dong et al., 2024). Moreover, the development of these approaches relies heavily on the domain knowledge and inductive biases of the researchers, thus requiring extensive trial and error experimentation. In addition, these heuristic approaches are susceptible to oversimplifications, often leading to locally sub-optimal solutions. Therefore, symbolic formula optimization (e.g., Dong et al. (2024); Chen et al. (2024); Ruan et al. (2024)) has been gaining ground as it explores the search space more efficiently using search algorithms. PrunerZero Dong et al. (2024) is one such post-training unstructured sparsity technique that employs a genetic algorithm to discover new symbolic formulas, outperforming manually crafted ones. Although genetic algorithms possess significant optimization capabilities, they focus primarily on the evolutionary process, but there's ample opportunity to enhance them by leveraging large language models, which have absorbed vast mathematical knowledge and intuitions, and by dynamically tuning their behavior through natural language instructions.

Given the paramount importance of designing an optimal pruning criterion, we ask whether it is possible to develop a transparent, well-reasoned, language-guided discovery process leveraging the remarkable abilities of state-of-the-art LLMs. Language, a meticulously structured and codified form of human communication, uniquely characterizes human evolution, facilitating the preservation and exchange of ideas. Emulating humans, LLMs trained on high-quality data can store a vast amount of scientific knowledge, endowing them with the ability to write high-quality code, solve complex reasoning problems, and utilize tools out-of-the-box or in a zero-shot manner (Dubey et al., 2024). The abilities of LLMs go far beyond standard information retrieval: they enable dynamic reasoning, which includes learning from evaluated examples Brown et al. (2020) and external context Lewis et al. (2020). With these capabilities, LLMs have already been successfully used to guide the discovery process in fields such as reward modeling Ma et al. (2024) and preference optimization Lu et al. (2024). Search algorithms, particularly those guided by LLMs, possess reasoning abilities along with mathematical and coding skills, rendering them indispensable for uncovering novel solutions and circumventing the local minima that frequently hinder heuristic methods.

To this end, we introduce LLM-Guider, a generic language-guided symbolic formula optimization framework aimed at discovering optimal pruning criteria for post-training unstructured sparsity in LLMs. LLM-Guider operates through three distinct yet interrelated stages: In the *Example Selection* stage, we identify the most promising k-shot seed formulas based on a predefined policy. The subsequent *Formula Generation* stage leverages the capabilities of LLMs, geared with customized hints and k-shot examples, to produce novel symbolic formulas tailored to our pruning objectives. Finally, in the *Formula Evaluation* stage, these generated formulas are rigorously assessed for their effectiveness in enhancing model sparsity while balancing the performance trade-off, with the highest-scoring formulas fed back into the generation pool for iterative refinement. Unlike genetic algorithms, this structured approach not only enables the dynamic exploration of the formula space but also allows for the integration of domain-specific insights through sampled hints, ensuring that our generated formulas are both novel and relevant. Our contributions are as follows:

- We propose LLM-Guider, a generic language-guided symbolic formula optimization framework, tailored for discovering optimal pruning criteria for post-training unstructured sparsity in LLMs. With minimal modifications, LLM-Guider can be re-oriented towards other applications that involve the structured discovery of symbolic formulas.

- We conduct extensive analysis on post-training unstructured sparsity benchmarks and show that formulas discovered through LLM-Guider outperform considered baselines.

- Unlike genetic algorithms, LLM-Guider enables the incorporation of domain and mathematical knowledge through hints and in-context examples. We also extend the aggregation strategies over the calibration dataset, resulting in never-seen-before novel pruning metrics.

## 2 Related Work

### 2.1 Sparsity in LLMs

Deep neural networks are typically dense and over-parameterized, leading to enormous computation and memory costs. Sparsity has emerged as a leading approach to the creation of more efficient models that function within high-dimensional feature spaces while simultaneously reducing representational complexity by utilizing only a subset of dimensions at a given time (Hoefler et al., 2021). There are two main forms of sparsity: structured and unstructured sparsity. Structured sparsity focuses on removing larger structures, which for LLMs includes layers Men et al. (2024), attention heads Venkataramanan et al. (2023), neurons, weight blocks, N:M-structures, embeddings, and hidden dimensions (Liu et al., 2023b; Xia et al., 2024; Zhou et al., 2021). On the other hand, unstructured sparsity prunes individual weights without regard to their structural grouping. Unstructured sparsity is usually associated with an importance matrix, which ranks the weights based on certain criteria. To compute concrete importance scores, a small calibration dataset drawn from the training distribution is first passed through the unpruned model; for each weight, we collect local statistics for activations and gradients, aggregate them using aggregation functions over the calibration set, and feed those aggregated features into the pruning formula under test—producing per-weight scores that are then sorted and thresholded to achieve the target sparsity.

Different unstructured sparsity algorithms vary in how they compute this weight importance matrix and which input parameters they depend on. Standard magnitude pruning (Cheng et al., 2024) uses the absolute weight magnitude for pruning decisions, based on the intuition that weights with smaller magnitudes contribute less to the network's output. However, the outcome of a neural network is not solely decided by the weight magnitudes. Even when a weight has a small magnitude, it can significantly contribute to the result if amplified by a large activation. To this end, Wanda (Sun et al., 2024) and RIA (Zhang et al., 2024) proposed an activation-based unstructured sparsity criterion leveraging the fact that output activations depend on both weight and input values. Additionally, methods like GBLM-Pruner (Das et al., 2024) and Pruner-Zero (Dong et al., 2024) incorporate gradients into the pruning decision, often outperforming activation-based methods. Large gradients indicate that the network is learning and is sensitive to parameter changes. SparseGPT (Frantar & Alistarh, 2023) goes one step further and uses the Hessian matrix approximation using activations, which is used both in pruning and optimal recovery. The key to designing effective unstructured sparsity algorithms lies in defining an appropriate weight importance formula/pruning metric. Research suggests that we can move beyond traditional approaches based solely on heuristics, instead leveraging a mixture of inputs to achieve optimal performance. This shift allows us to redefine the process as a search over possible formulas, enabling a more nuanced and data-driven approach to weight importance.

"'latex

### 2.2 Symbolic Formula Optimization

While there is some clarity regarding what an effective sparsity formula should depend on, the specific symbolic formula remains an area of active research. Most works focus on manually crafting weight importance formulas based on heuristics (Dong et al., 2024). However, these heuristic approaches are prone to oversimplifications and can result in locally sub-optimal solutions. Therefore, it is a common practice to use global model-based approaches that explore the search space efficiently using search algorithms. For example, PrunerZero (Dong et al., 2024) employs genetic algorithms to discover unstructured sparsity formulas, which have outperformed existing heuristic-based methods. Similarly, the successful application of genetic algorithms led to the discovery of the Lion optimizer (Chen et al., 2023b). Reinforcement learning has also been used to discover novel neural network architectures (Zoph & Le, 2017) and activation functions (Ramachandran et al., 2017). With recent breakthroughs in language modelling, LLMs have been readily used for language-guided search processes. For instance, LLM-guided search found code functions for reinforcement learning rewards in Eureka (Ma et al., 2024). A similar approach was also used to discover novel alignment formulas in (Lu et al., 2024).

However, existing symbolic pruning methods remain limited in how they explore and control the formula space. In particular, PrunerZero relies on a genetic algorithm operating over a fixed, manually specified

grammar. While this design enables systematic exploration, it restricts the search to formulas expressible within the predefined symbolic space and typically requires substantial random exploration before high-performing regions are identified. Moreover, modifying the search behavior generally requires changing the grammar, evolutionary operators, or search configuration.

LLM-Guider builds on the broader idea of symbolic formula discovery but differs from grammar-based evolutionary methods by performing prompt-conditioned code generation. This allows candidate pruning formulas to be shaped by natural-language instructions, k-shot examples, and domain-specific hints. As a result, the search space can be adapted at inference time without redesigning the underlying optimization algorithm. The LLM can combine weight, activation, and gradient statistics using normalization terms, stabilizing constants, and operation patterns that are not necessarily hard-coded in a fixed grammar.

This provides LLM-Guider with three key advantages over prior symbolic pruning approaches. First, it enables more flexible exploration beyond a manually predefined grammar. Second, it allows human intuition and domain knowledge to be injected directly through prompts, enabling the search to emphasize interpretability, diversity, stability, or variable constraints. Third, it leverages the LLM's prior mathematical and programming knowledge to guide formula generation from the beginning, rather than relying primarily on random initialization and evolutionary refinement. Therefore, LLM-Guider differs from prior symbolic pruning approaches not only in its search mechanism, but also in its flexibility, controllability, and ability to incorporate domain knowledge during formula discovery. "'

## 2.3 Prompting

LLMs effectively learn the nuances of token distributions, significantly improving their capability to tackle a diverse array of tasks during training (Wei et al., 2022a). However, to fully optimize LLM performance, it is necessary to use inference-time optimization techniques, applied after the model has been trained. Inference-time optimization involves both enhancing the prompt itself and employing techniques that solve problems using a multi-step approach. Role prompts (Kong et al., 2024) allow the LLM to adjust its style and focus on specific target tasks. Additionally, *K-shot prompting* (Brown et al., 2020) enables models to learn from in-context information and generate output based on examples. Progressive hints in prompts (Zheng et al., 2024) have also been shown to enhance the reasoning abilities of LLMs. A common practice is to decompose problems into a *Chain of Thought (CoT)* (Wei et al., 2022b), which allows the model to solve complex tasks step-by-step. Furthermore, the *Reflection* approach (Schulhoff et al., 2024) is often used to assess and improve generated responses. To go beyond the sequential nature of CoT, the *Tree of Thoughts (ToT)* (Yao et al., 2023) introduces a combination of parallel and sequential generation. To optimize LLM usage during inference, it is beneficial to enhance prompts with hints and cues for additional in-context information. Single-turn prompting underperforms compared to multi-step approaches, which effectively decompose prompts and verify each step's outcome. In conclusion, complex tasks require advanced prompting techniques for optimal results.

## 3 Methodology

LLM-Guider is a generic symbolic formula optimization framework that operates through three distinct stages: k-shot example selection, formula generation, and evaluation. It leverages the capabilities of a large language model (LLM) to efficiently explore the formula space. In the subsequent subsections, we provide detailed descriptions of each component within our framework.

### 3.1 Example selection

### 3.1.1 Search Space Design

The search space design serves as the foundation for LLM-guided formula discovery. The search space is composed of weights, activations, and gradients, combined with operations that define the pruning metrics. Weight magnitudes are derived from the target LLM while activations and gradients are aggregated over a small calibration dataset to capture the required input statistics. Table 1 provides an overview of an

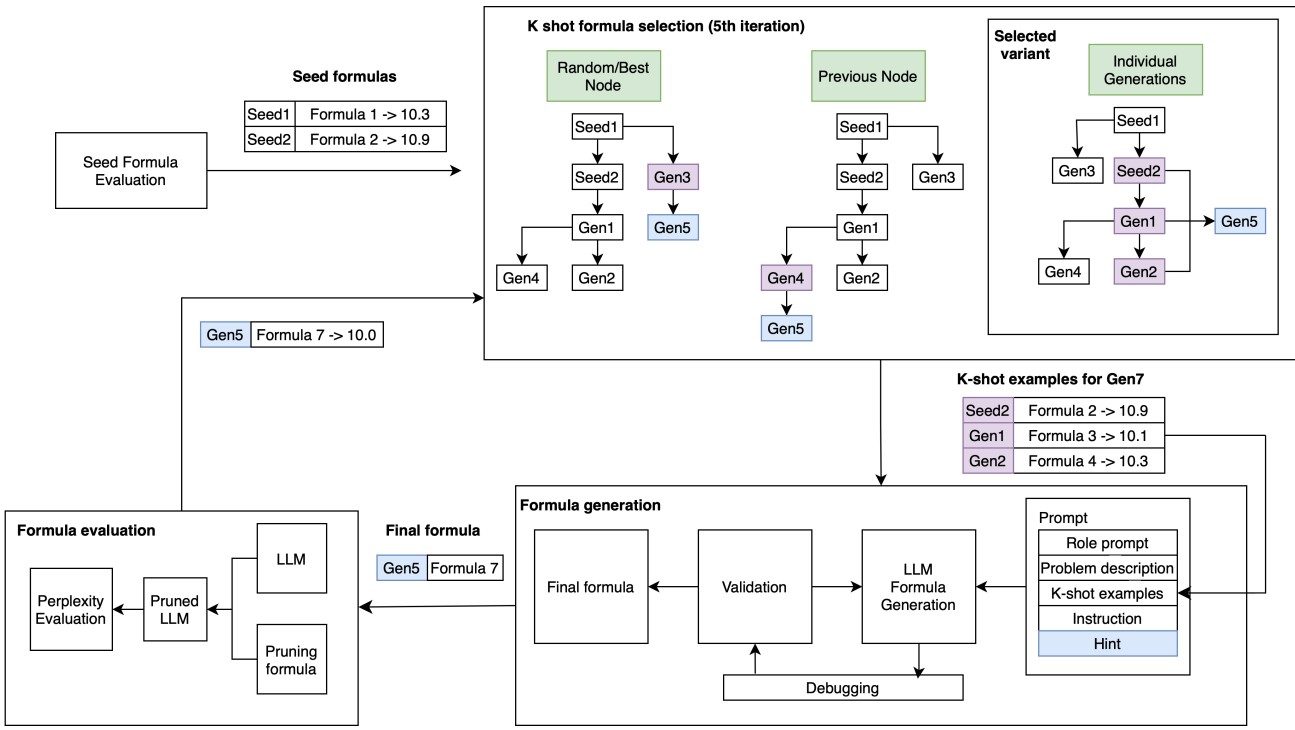

Figure 1: Overview of LLM-Guider, a generic framework for symbolic formula optimization, tailored for discovering optimal pruning criteria for post-training unstructured sparsity in LLMs: In the *Example Selection* stage, we identify promising k-shot seed formulas based on a predefined policy. Here, a node denotes an intermediate candidate formula in the search tree, and its associated floating-point score represents the evaluation value used to rank candidate formulas. The k-shot formula selection step chooses compact, high-performing formulas from previously evaluated candidates, encouraging both effectiveness and interpretability. The subsequent *Formula Generation* stage leverages the capabilities of LLMs, geared with customized hints and k-shot examples, to produce novel symbolic formulas tailored to our pruning objectives. In the *Formula Evaluation* stage, these generated formulas are rigorously assessed, and the highest-scoring formulas are fed back into the generation pool for iterative refinement.

extended list of aggregation strategies over the calibration dataset. LLM-Guider treats these strategies as input variables and autonomously determines which operations to use when generating the symbolic formulas through iterative refinement. Unless constrained by hints, the operations over these input variables are not set explicitly.

The pruning metric defines the importance of weights in a model, determining which are retained or pruned under a predefined sparsity threshold. These formulas are represented as code with a predefined header, ensuring seamless integration and execution during the evaluation process. This coding approach also mitigates format conversion issues, promoting consistency and efficiency.

| Category | Names |
|---|---|
| Weights | $W$ |
| Activations | $A_{\mathrm{mean}}$, $A_{M^2}$, $A_{\mathrm{sum\_squares}}$, $A_{\mathrm{sum\_abs}}$, $A_{\mathrm{min}}$, $A_{\mathrm{max}}$, $A_{\mathrm{mean\_abs}}$, $A_{\mathrm{mean\_squared}}$, $A_{\mathrm{variance}}$, $A_{\mathrm{std}}$ |
| Gradients | $G_{\mathrm{mean}}$, $G_{\mathrm{L1}}$, $G_{\mathrm{L2}}$, $G_{M^2}$, $G_{\mathrm{sum\_gradients}}$, $G_{\mathrm{sum\_abs\_gradients}}$, $G_{\mathrm{sum\_gradients\_squared}}$, $G_{\mathrm{mean\_gradients}}$, $G_{\mathrm{mean\_abs\_gradients}}$, $G_{\mathrm{mean\_gradients\_squared}}$, $G_{\mathrm{variance}}$, $G_{\mathrm{std}}$ |

Table 1: An exhaustive list of input variables employed in the search space design of LLM-Guider. More explanation on these can be found in Appendix E.

### 3.1.2 K-Shot Example Selection

Seed formulas are essential for initializing the pruning metric discovery process. They provide a structured foundation for subsequent LLM-guided exploration by defining initial examples. Seed formulas are provided upfront as code, including classical magnitude pruning and custom formulas that combine weights, activations, and gradients. These formulas are evaluated to obtain initial performance scores. By serving as a reliable starting point, seed formulas guide LLMs to generate meaningful pruning metrics and uphold a consistent response format, like the role of correct examples in few-shot learning.

K-shot examples are an integral component of the LLM-guided framework. As they are included in the prompt, they enable in-context learning, supporting reasoning and iterative improvement. These examples form the foundation of each generation, helping the model build upon past successes and avoid previous failures. Each generation relies on k-shot examples, as described in Liu et al. (2023a), to enhance performance and draw conclusions based on previously evaluated attempts. Specifically:

- **Generation References:** Each new generation references examples from the pool of past attempts.

- **Evaluation Scores:** Examples are paired with their evaluation scores, providing clear insights into what strategies worked and which failed.

- **Selection Strategies:** The process for selecting k-examples includes:
  - Randomly selecting and expanding past the node
  - Selecting the best node
  - Using the node from the previous evaluation and iteratively expanding it
  - Selecting the top-n individual generations with the best scores to form a context.

Initially, no prior generations exist to select as examples. Following existing work (Chen et al., 2024), an initial pool of predefined examples is created and evaluated. In our approach, we employ a single Wanda (Sun et al., 2024) seed formula that simultaneously integrates both weights and activations.

Through carefully managed k-shot example selection, the framework achieves a balance between exploration of new possibilities and refinement of high-performing approaches. This balance ensures efficient and effective formula discovery, leveraging prior knowledge while encouraging innovation.

## 3.2 LLM-Based Symbolic Formula Generation

Unstructured sparsity operates by applying a mask over weights. To enhance this process, an LLM is employed to generate new weight importance matrices through prompting. The LLM leverages prior knowledge and dynamic reasoning to create novel sparsity formulas.

The prompts used to guide the LLM consist of the following structured parts:

- **Role Prompt:** Defines the task's context to focus the LLM on pruning objectives.

- **Instruction Prompt:** Provides formula objectives, emphasizing interpretability and detailing available variables (e.g., weights, activations, and gradients aggregations). For each variable, the size of its corresponding tensor is explicitly provided, ensuring the LLM can appropriately handle and process the input data.

- **k-Shot Examples:** Supplies selected past evaluations, enabling reasoning over previously successful attempts.

- **Hints:** Offers domain-specific guidance such as normalization or variable constraints. These are sampled from a predefined pool using strategies like uniform or weighted sampling and are included as textual parts of the prompt.

The LLM combines its internal knowledge with external hints and reasoning over k-shot examples to dynamically generate innovative and effective sparsity formulas. By integrating structured prompts and leveraging both static and dynamic knowledge sources, the LLM serves as a central tool for discovering novel sparsity formulas. This approach ensures adaptability and precision in tailoring pruning strategies to specific tasks.

When symbolic formulas are generated in natural language by the LLM, there is no guarantee that the resulting code will compile or execute correctly. This introduces a need for a robust validation and refinement process to ensure the correctness of the formulas. Validation begins by running the generated formulas in an evaluation environment. If execution fails, a debugging phase is triggered.

The LLM mimics human debugging by verifying tensor sizes step-by-step. It augments the initial code with a detailed walkthrough of tensor sizes to identify inconsistencies or unsupported operations. Once inconsistencies are detected, the LLM attempts to fix the issues using the gathered size information. This process is repeated for a predefined number of attempts to validate and correct the formulas efficiently.

The iterative validation and refinement process ensures the correctness of LLM-generated formulas. By systematically identifying and resolving errors, this approach guarantees reliable symbolic pruning metrics, even when initially created through natural language.

### 3.3 Formula Evaluation

Having generated a candidate formula, the next step is to evaluate its performance so that LLM-Guider can quantify its effectiveness.

In our framework, we execute the evaluation in three sequential steps. We begin by applying the selected symbolic formula to our precomputed statistics on weights, activations, and gradients, which yields a binary mask indicating which parameters to keep and which to remove. Using this mask, we prune the language model by zeroing out the designated weights, producing a leaner version of the network without any additional fine-tuning. Finally, we assess the pruned model's quality by running it on the WikiText-2 test set and recording its perplexity.

Once a formula has been evaluated and its performance recorded, it is added to the k-shot example pool for the next generation. This cycle of formula generation, evaluation, and example selection repeats until the predefined number of iterations is completed.

## 4 Discovered Formulas

Our framework introduces novel pruning metrics derived through extensive experimentation on SmolLM2 model, described in detail in Appendix A. Specifically, it discovered two effective pruning formulas:

1. **Best-Performing Formula:** This metric combines normalization of activations, average gradients, and gradient variability, hypothesizing that parameters with higher gradient variability play critical roles in optimization. It was found by modifying LLM-Guider baseline configuration with diversity hints

$$I = W \odot \left[ \left( \frac{A_{\mathrm{mean}} - A_{\min}}{A_{\max} - A_{\min} + \epsilon} \right) G_{\mathrm{mean}}^{\top} + G_{\mathrm{mean\_abs}} \right] \odot G_{\mathrm{std}}$$

2. **Second-Best Formula:** This formulation emphasizes gradient variability weighted by parameter magnitudes, capturing critical gradient variations essential for robust generalization across models.

$$I = |W| \odot G_{\mathrm{std}} \tag{1}$$

Traditional pruning methods typically rely on mean- or magnitude-based norms of activations and gradients, potentially overlooking parameters exhibiting small but significant variability. In contrast, our proposed metrics explicitly incorporate statistical aggregates such as gradient standard deviation and activation variability, capturing parameter importance more effectively. However, it is important to note that metrics specifically tailored to individual models may risk overfitting, thereby diminishing their generalization capabilities.

The LLM also decided to include a small $\epsilon$ term to avoid divide-by-zero issues ($\epsilon = 10^{-12}$). This was not hard-coded: it was introduced as part of its reasoning about numerical stability. This suggests the model was not only searching for better scoring functions, but also self-correcting them for safe numerical behavior.

Interestingly, the best-performing formula can be interpreted as a generalized version of the second-best-performing one, as it adds an additional factor to the multiplication. This observation suggests that, while the LLM was capable of discovering a formula with a simple structure, it also demonstrated the ability to refine and extend it into a more complex and effective form.

## 5 Experiments

### 5.1 Evaluation Setup

We evaluate the effectiveness of LLM-guided search using a single model family. Specifically, we employ SmolLM2-135M Allal et al. (2025), a lightweight language model designed for computationally efficient experimentation. We use GPT-4o mini for formula generation. Model performance is assessed across two primary benchmarks. For language modeling, we report perplexity on the WikiText2 test set Merity et al. (2016). For zero-shot generalization, we evaluate using EleutherAI's LM Harness framework, which includes a diverse set of tasks: ARC Challenge (Clark et al., 2018), ARC Easy (Clark et al., 2018), BoolQ (Clark et al., 2019), OpenBookQA (Mihaylov et al., 2018), RTE (Wang et al., 2019), Winogrande (Sakaguchi et al., 2021), and HellaSwag (Zellers et al., 2019).

LLM-Guider is designed to search for optimal symbolic formulas, with 100 generations evaluated per run. Our empirical studies, using greedy search, led to the framework configuration in Appendix G. The process takes approximately 1.5 hours on a single A100 GPU.

To compute pruning metrics at each iteration, we rely on weights, activations, and gradients. Multiple statistical measures for these components are precomputed, as detailed in Appendix E. We used a fixed set of 128 calibration samples to precompute statistics, following the Wanda approach Sun et al. (2024), which ensures that the unstructured sparsity stabilizes at optimal levels. During each iteration, an importance matrix is computed on the basis of these statistics, and pruning is applied based on a sparsity ratio of 0.5. Similarly to previous work, we use the first fragment of the C4 dataset Raffel et al. (2020) for the evaluation.

### 5.2 Baselines

We evaluated the performance of our approach against several established pruning methods, each taking advantage of different combinations of weights, activations, and gradients to compute pruning metrics. Specifically, we compare with standard magnitude pruning, Wanda (Sun et al., 2024), which incorporates both weights and activations; PrunerZero (Dong et al., 2024), a state-of-the-art method based on weights and gradients; and SparseGPT (Frantar & Alistarh, 2023), which uses Hessian information and error propagation to update weights. We also include the dense (unpruned) model corresponding to a sparsity ratio of 0 as an additional baseline. These diverse baselines serve as strong reference points, allowing us to assess the effectiveness of our LLM-guided search in achieving greater sparsity while preserving competitive model performance.

### 5.3 Language Modeling

Language modeling using perplexity allows for assessing how well a language model predicts a given sequence of text. Lower perplexity indicates that the model assigns higher probabilities to the correct words, meaning that it has a better understanding of the language and generates a more fluent, coherent text.

Based on the results in Table 2, LLM-Guider outperforms all other methods that do not require weight updates. Furthermore, it significantly narrows the performance gap with SparseGPT, the only algorithm that utilizes weight updates. This trend aligns with findings from the Pruner-Zero experiment conducted on OPT models, which showed that SparseGPT remains competitive with Pruner-Zero due to its ability to

update weights. Notably, the same experiment also demonstrated that the performance difference between these methods becomes more pronounced in models with smaller parameter counts.

| Method | Weight Update | Perplexity ↓ |
|---|---|---|
| LLM-Guider | 55 | **30.99** |
| Magnitude | 55 | 536.44 |
| Wanda | 55 | 31.66 |
| Pruner-Zero | 55 | 33.09 |
| SparseGPT | 51 | **30.83** |

Table 2: Comparison to state-of-the-art methods for SmolLM2-135M using WikiText-2 perplexity with sparsity ratio 0.5

| Method | arc_challenge | arc_easy | boolq | hellaswag | openbookqa | rte | winogrande | Mean ↑ |
|---|---|---|---|---|---|---|---|---|
| Dense | $26.11 \pm 1.28$ | $54.12 \pm 1.02$ | $42.91 \pm 0.87$ | $34.94 \pm 0.48$ | $22.00 \pm 1.85$ | $51.26 \pm 3.01$ | $51.62 \pm 1.40$ | $40.42 \pm 1.42$ |
| LLM Guider (Best) | $20.73 \pm 1.18$ | $44.07 \pm 1.02$ | **$62.42 \pm 0.85$** | $29.82 \pm 0.46$ | $16.80 \pm 1.67$ | **$57.04 \pm 2.98$** | **$51.22 \pm 1.40$** | **$40.30 \pm 1.37$** |
| LLM Guider (Second) | **$21.08 \pm 1.19$** | $43.90 \pm 1.02$ | $62.35 \pm 0.85$ | $29.82 \pm 0.46$ | $16.40 \pm 1.66$ | $56.32 \pm 2.99$ | **$51.22 \pm 1.40$** | $40.16 \pm 1.37$ |
| PrunerZero | $19.03 \pm 1.15$ | **$44.19 \pm 1.02$** | $56.73 \pm 0.87$ | $29.81 \pm 0.46$ | $15.60 \pm 1.62$ | $54.51 \pm 3.00$ | $49.88 \pm 1.41$ | $38.54 \pm 1.36$ |
| Wanda | $20.65 \pm 1.18$ | $43.10 \pm 1.02$ | $60.92 \pm 0.85$ | $29.94 \pm 0.46$ | $15.20 \pm 1.61$ | $51.26 \pm 3.01$ | $51.14 \pm 1.40$ | $38.89 \pm 1.36$ |
| SparseGPT | $20.82 \pm 1.19$ | $41.67 \pm 1.01$ | $57.89 \pm 0.86$ | **$30.56 \pm 0.46$** | **$17.00 \pm 1.68$** | $52.71 \pm 3.01$ | $50.83 \pm 1.41$ | $38.78 \pm 1.37$ |
| Magnitude | $19.37 \pm 1.15$ | $35.56 \pm 0.98$ | $38.01 \pm 0.85$ | $26.78 \pm 0.44$ | $13.20 \pm 1.52$ | $54.51 \pm 3.00$ | $50.59 \pm 1.41$ | $34.00 \pm 1.34$ |

Table 3: Accuracies (%) of SmolLM2-135M for 7 zero-shot tasks with unstructured 50% sparsity.

### 5.4 Zero-Shot Evaluation

We conducted extensive experiments to evaluate our model across a comprehensive suite of zero-shot commonsense reasoning tasks. As detailed in Table 3, evaluation performance varied considerably across tasks. Notably, on benchmarks such as BoolQ, RTE, and WinoGrande, our method demonstrated a clear advantage over baseline approaches. With an overall mean accuracy of 40.30%, our approach significantly surpasses the Wanda baseline (38.89%) and compares favorably with the Dense model (40.42%). These findings underscore that pruning based on a calibration dataset yields robust improvements in downstream performance.

## 6 Conclusions

We introduced LLM-Guider, a language-guided symbolic formula optimization framework designed to discover novel pruning metrics for post-training unstructured sparsity in large language models. Our approach leverages the advanced reasoning and coding capabilities of modern LLMs by combining domain-specific hints, k-shot examples, and iterative refinement to generate and validate effective symbolic formulas.

Through extensive experiments on SmolLM2-135M, we demonstrated that the formulas discovered by LLM-Guider outperform traditional methods such as magnitude pruning, Wanda, and PrunerZero—achieving lower perplexity and competitive zero-shot performance without requiring weight updates. Detailed ablation studies further highlighted the impact of key components such as seed formula selection, generation strategy, and tailored hint configurations, confirming that even minimal human guidance can significantly enhance the discovery process.

Our best-performing formula, which integrates normalized activation statistics, average gradients, and gradient variability, underscores the benefit of incorporating richer statistical aggregates beyond standard mean-based approaches. Overall, LLM-Guider not only advances the state-of-the-art in unstructured sparsity but also establishes a transparent and systematic methodology for symbolic optimization in neural networks.

## Limitations

With extensive experiments and analysis, we show that LLM-Guider framework can successfully find a state-of-the-art solution in the SmolLM2-135M model at 50% sparsity. This highlights the potential of language guided search for automating sparsity-aware optimization. Applying our method to different model sizes or pruning levels may require re-running the framework to find an optimal solution tailored to each specific configuration. Future work could explore what factors enable solutions to transfer across models and sparsity levels, helping reduce the need for full recalibration.

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

# A   Ablations

LLM-Guider framework consists of multiple variable parts, including seed formula, k-shot generation strategy, role prompt, and hints. In this section, we run multiple ablations on the baseline to find the most impactful decisions. Each of the ablations is run for 100 generations, using 128 calibration samples with a sparsity ratio of 0.5 and SmolLM2-135M-Instruct Allal et al. (2025) as the model. Each generation denotes one complete selection–generation–evaluation loop. We repeat the experiment over 3 seeds.

First, we begin by investigating the effect of different seed formulas. We consider rules that utilize weights, activations, and gradients. Empirically, we have found that the activation-based metric, Wanda, outperforms other options. The most unstable runs occur when using the magnitude formula alone. Our results from Table 4 indicate that formulas combining at least two of the three components—weights, activations, and gradients—significantly outperform using magnitude alone. Additionally, incorporating PrunerZero into the magnitude formula leads to notable improvements in evaluation metrics. Among all the metrics tested, Wanda is mathematically the most complex, as it involves matrix and row multiplications, whereas other approaches rely on simpler element-wise operations. We hypothesize that this added complexity enables the framework to generate more effective formulas.

| Seed formulas | Best Perplexity | Mean Perplexity |
|---|---|---|
| Wanda (baseline) | **31.02** | **31.74** $_{\pm 1.01}$ |
| Magnitude + PrunerZero | **31.02** | 32.69 $_{\pm 2.89}$ |
| PrunerZero | 31.30 | 35.10 $_{\pm 6.59}$ |
| Magnitude + Wanda + PrunerZero | 31.03 | 39.58 $_{\pm 7.49}$ |
| Magnitude | 31.30 | 82.67 $_{\pm 85.82}$ |

Table 4: Comparison of seed formulas for LLM-Guider init on SmolLM2-135M (WikiText-2, sparsity 0.5)

We analyze the impact of different generation strategies based on Table 5. Specifically, we evaluate best node selection, random node selection, selection of top individual generations, and a uniform strategy that combines best, random, and previous nodes. An effective generation strategy should allow the LLM to explore both high-quality and diverse solutions. Our results indicate that, under the tested conditions, diversity plays a crucial role. Notably, random node generation slightly outperforms the best node selection strategy, highlighting the importance of exploration in addition to exploitation.

| Generation Strategy | best_metric | mean_metric |
|---|---|---|
| Best Node (baseline) | 31.02 | 31.74 $_{\pm 1.01}$ |
| Random Node | 31.02 | **31.54** $_{\pm 0.90}$ |
| Top 8 Individual Generations | 31.02 | 35.73 $_{\pm 8.17}$ |
| Uniform (Best, Random, Previous Node) | 31.50 | 33.71 $_{\pm 2.25}$ |

Table 5: Comparison of generation strategies

Role prompting is the technique to tune style of the response. In our experiments from Table 6 we have found that using role prompt did not improve generation results. We hypothesize that style of response is not that important for task of unstructured sparsity formula generation.

| Use role prompt | Best Perplexity | Mean Perplexity |
|---|---|---|
| 51 (baseline) | **31.02** | 31.74 $_{\pm 1.01}$ |
| 55 | **31.02** | **31.44** $_{\pm 0.74}$ |

Table 6: Ablation on role prompt

Hints are part of the framework that allows for the most flexibility and enhancements. As observed in Table 7, in our experimental regime, the strongest results were achieved by limiting the number of variables that

the LLM uses. When it comes to top-generation diversity, only hints allowed improvements over the best metrics; however, they were unstable in repetitive runs. This indicates that the usage of limiting hints was critical for stability, whereas adding hints that promote variety was beneficial for improving the best metrics. Notably, the selection of hints focusing on complexity or employing complex operations, such as matrix multiplication alone, performed the poorest.

| Hint Configuration | Best Perplexity | Mean Perplexity |
|---|---|---|
| Baseline | 31.02 | 31.74 ± 1.01 |
| Diversity Hints | **30.99** | 36.54 ± 5.46 |
| Limiting Hints | 31.02 | **31.21** ± 0.16 |
| Domain-specific | 32.04 | 32.54 ± 0.48 |
| Inspiration Hints | 34.31 | 35.31 ± 1.20 |
| Reflection Hints | 32.70 | 36.60 ± 6.25 |
| Diversity and Limiting | 31.30 | 34.31 ± 2.76 |
| Reflection and Limiting | 31.30 | 35.75 ± 7.71 |
| Reflection and Domain-specific | 31.02 | 36.19 ± 5.48 |
| Complexity Hints | 31.30 | 168.14 ± 234.32 |
| Matrix Multiplication Hints | 36.93 | 288.08 ± 217.70 |
| Matrix Operations Correctness Hints | 41.93 | 203.53 ± 273.89 |

Table 7: Ablation comparing effectiveness of different hint configurations

We studied the impact of retries during the debugging phase. This experiment demonstrates the effectiveness of debugging in improving formula generation. Our results, as shown in Table 8, indicate that including debugging retries enhances formula generation. This is a natural conclusion, as it leads to a higher number of correctly generated formulas. However, in our study, the impact was particularly visible when up to five retries were performed for each generation.

| Number of Retries | Best Perplexity | Mean Perplexity |
|---|---|---|
| 0 | 31.02 | 31.38 ± 0.41 |
| 2 (baseline) | 31.02 | 31.74 ± 1.01 |
| 5 | 31.02 | **31.21** ± 0.16 |

Table 8: Performance comparison on number of retries during debugging

We evaluate the framework without each prompt component in isolation and jointly, and we present results in Table 9. This systematic analysis confirms that both hints and k-shot examples play pivotal roles in guiding the LLM toward performant and diverse symbolic formulas. In particular, hints improve *best-case* perplexity by expanding the exploratory radius of the search, whereas k-shot examples improve *average-case* perplexity by stabilizing convergence. Taken together, these results validate that static guidance (hints) and in-context learning (examples) are synergistic in navigating the symbolic formula space.

| Configuration | Best Perplexity ↓ | Mean Perplexity ↓ |
|---|---|---|
| LLM-Guider (full) | **31.02** | **31.74** ± 1.01 |
| No hints | 34.98 | 37.21 ± 2.99 |
| No k-shot examples | 32.58 | 34.01 ± 1.54 |
| No hints & no k-shot | 35.64 | 38.88 ± 5.26 |

Table 9: Ablation over prompt scaffolding on SmolLM2-135M (WikiText-2, sparsity 0.5). Removing hints, k-shot examples, or both.

## B  Cross-Model Generalization

To assess generalization beyond SmolLM2-135M, we evaluated the top-performing LLM-Guider formulas on the MobileLLM-125M Liu et al. (2024) model. Despite architectural differences, these formulas reduced

perplexity relative to pruning baselines while preserving zero-shot accuracy across downstream tasks. This suggests that the discovered formulas capture generalizable statistical patterns in weight–activation–gradient interactions.

| Method | PPL ↓ | ARC-C | ARC-E | BoolQ | Hella. | OBQA | RTE | Wino | Mean ↑ |
|---|---|---|---|---|---|---|---|---|---|
| Dense | 12.53 | 0.20 | 0.46 | 0.60 | 0.33 | 0.18 | 0.54 | 0.52 | 0.40 |
| PrunerZero | 25.59 | 0.20 | 0.36 | 0.62 | 0.29 | 0.16 | 0.53 | 0.52 | 0.38 |
| Wanda | 28.57 | 0.18 | 0.37 | 0.62 | 0.29 | 0.15 | 0.51 | 0.51 | 0.38 |
| LLM-Guider (best) | 24.80 | 0.20 | 0.37 | 0.62 | 0.29 | 0.14 | 0.52 | 0.52 | 0.38 |
| LLM-Guider (second) | 24.65 | 0.20 | 0.38 | 0.62 | 0.29 | 0.15 | 0.52 | 0.51 | 0.38 |

Table 10: MobileLLM-125M evaluation. LLM-Guider formulas maintain or improve perplexity relative to standard pruning baselines while preserving zero-shot performance.

This evaluation supports the hypothesis that LLM-Guider yields pruning formulas that generalize across model families and tasks.

## C  Model Family GeneralizationModel Family Generalization

We evaluate whether pruning formulas discovered on a smaller model transfer to a larger model family. Specifically, we compare the best and second-best LLM-Guider formulas discovered on SmolLM2-135M against PrunerZero when applied to LLaMA-2-7B. The transferred LLM-Guider formulas achieve competitive performance with PrunerZero, suggesting that the discovered gradient-based pruning metrics capture patterns that generalize beyond the model used during search.

However, running LLM-Guider directly on the target LLaMA-2-7B architecture further improves zero-shot performance over PrunerZero, as shown in Figure 2.. This indicates that while formulas discovered on smaller models can transfer reasonably well, optimal pruning performance may require a dedicated search on the target model architecture. These results highlight both the transferability of LLM-Guider-discovered formulas and the benefit of model-specific symbolic formula optimization.

## D  Search Efficiency Compared to PrunerZero

We additionally compare the search efficiency of LLM-Guider against PrunerZero. PrunerZero uses a genetic algorithm and requires a substantial initial exploration phase: it starts with 300 randomly generated formulas and then performs evolutionary search over 150 epochs. This design enables broad exploration, but it also requires many candidate formulas to be sampled before the search can exploit high-performing regions of the symbolic space.

LLM-Guider follows a different strategy. Instead of relying on a large random initialization pool, it starts from a single seed formula and uses the LLM's prior knowledge, in-context examples, and sampled hints to guide formula generation. In our main experiments, LLM-Guider is run for 100 generations. Moreover, in an example run, we observed that the search can saturate much earlier, with the best formula emerging within approximately 20 generations. This suggests that language-guided search can reduce the number of required evaluations by quickly concentrating on promising regions of the formula space.

## E  Inputs

Below, we provide an extensive list of variables used in LLM-Guider. These variables collectively define the search space for the LLM to find an optimal pruning metric that effectively sparsifies the target LLM with minimal loss in performance.

**Weights**  $W$: Weights of the model.

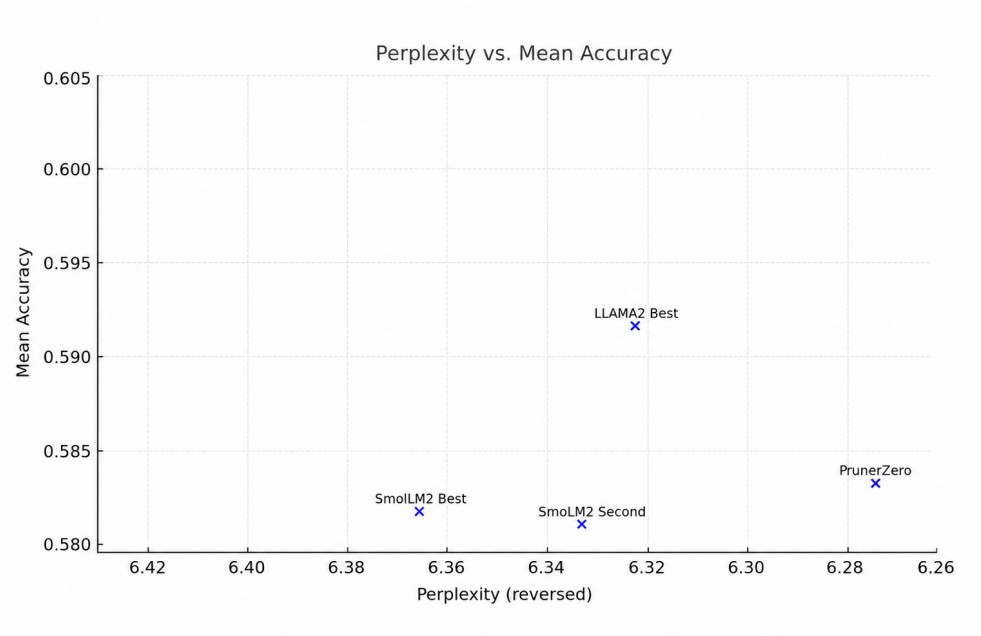

Figure 2: Comparison of LLM-Guider formulas discovered on SmolLM2-135M and transferred to LLaMA-2-7B against a formula discovered by running LLM-Guider directly on LLaMA-2-7B, using a sparsity ratio of 0.5. The transferred formulas remain competitive with PrunerZero, while the LLaMA-2-7B-specific search further improves zero-shot performance.

**Activations**

- $A_{\text{mean}}$: Mean of the activations across batches.

- $A_{M^2}$: Accumulated sum of squared differences from the mean (used for calculating variance).

- $A_{\text{sum\_squares}}$: Sum of squares of the activations along the batch dimension.

- $A_{\text{sum\_abs}}$: Sum of absolute values of the activations along the batch dimension.

- $A_{\text{min}}$: Minimum value of the activations across batches.

- $A_{\text{max}}$: Maximum value of the activations across batches.

- $A_{\text{mean\_abs}}$: Mean of the absolute values of the activations along the batch dimension.

- $A_{\text{mean\_squared}}$: Mean of the squared values of the activations along the batch dimension.

- $A_{\text{variance}}$: Variance of the activations, computed from the accumulated sum of squared differences $M^2$.

- $A_{\text{std}}$: Standard deviation of the activations, computed as the square root of the variance.

**Gradients**

- $G_{\text{mean}}$: Mean of the gradients across batches.

- $G_{\text{L1}}$: L1 norm of the gradients across batches.

- $G_{\text{L2}}$: L2 norm of the gradients across batches.

- $G_{M^2}$: Accumulated sum of squared differences from the mean (used for calculating variance).

- $G_{\text{sum\_gradients}}$: Sum of gradients along the batch dimension.

- $G_{\text{sum\_abs\_gradients}}$: Sum of absolute values of the gradients along the batch dimension.

- $G_{\text{sum\_gradients\_squared}}$: Sum of squares of the gradients along the batch dimension.

- $G_{\text{mean\_gradients}}$: Mean of the gradients along the batch dimension.

- $G_{\text{mean\_abs\_gradients}}$: Mean of the absolute values of the gradients along the batch dimension.

- $G_{\text{mean\_gradients\_squared}}$: Mean of the squared gradients along the batch dimension.

- $G_{\text{variance}}$: Variance of the gradients, computed from the accumulated sum of squared differences $M^2$.

- $G_{\text{std}}$: Standard deviation of the gradients, computed as the square root of the variance.

## F   Baseline Pruning Metrics

### 1. Magnitude Pruning

The pruning score for each weight is
$$S_{ij} = |W_{ij}|$$
where

- $W_{ij}$ is the weight of the connection from neuron $j$ (input) to neuron $i$ (output).

### 2. SparseGPT

SparseGPT approximates the influence of each weight via the inverse Hessian:

$$S_{ij} \;=\; \frac{W_{ij}^2}{\left[(H^{-1})\right]_{ij}}$$

where

- $H$ is the (approximate) Hessian matrix of the loss w.r.t. weights.

- $(H^{-1})$ denotes the diagonal of the inverse Hessian.

### 3. Wanda

Wanda scores combine weight magnitude with activation norm:

$$S_{ij} \;=\; |W_{ij}| \,\times\, \|X_j\|_2$$

where

- $X_j \in \mathbb{R}^N$ is the vector of activations at neuron $j$ over a calibration dataset.

- $\|\cdot\|_2$ denotes the Euclidean norm.

**4. PrunerZero**

PrunerZero combines squared magnitude with scaled gradient magnitude:

$$S_{ij} = W_{ij}^2 \times \sigma(|G_{ij}|), \qquad \sigma(x) = \frac{x - \min(x)}{\max(x) - \min(x)}$$

where

- $G_{ij} = \frac{\partial \mathcal{L}}{\partial W_{ij}}$ is the gradient of the loss $\mathcal{L}$ w.r.t. $W_{ij}$.

- $\sigma(\cdot)$ denotes min–max normalization applied across all absolute gradient values.

## G    LLM-Guider Baseline

| Configuration Details | |
|---|---|
| **Number of Rounds** | 100 |
| **Number of Retries** | 2 |
| **Eureka Seed** | 0 |
| **Temperature** | 1 |
| **Use Role Prompt** | true |
| **K Examples** | wanda |
| **Sparsity Ratio** | 0.5 |
| **Number of Samples** | 128 |
| **Model** | HuggingFaceTB/SmolLM2-135M-Instruct |
| **Evaluator Seed** | 0 |
| **Hint Sampler Type** | UniformSampler |
| **Generation Strategy Sampler Type** | UniformGenerationStrategySampler |
| **Generation Strategy Value Type** | BestNodeStrategy |
| **Hint Options** | LimitVariablesHint (value: 2) |
| | LimitVariablesHint (value: 3) |
| | LimitVariablesHint (value: 4) |
| | ComplementMatchingSizeHint |
| | UnaryOperationsHint |
| | TryDifferentHint |
| | AlternativePerspectiveHint |

Table 11: Configuration Table

## H    Hints

Hints Overview: This document provides a summary of the various hint types used to guide a problem-solving process. The hints are organized into several categories that serve distinct purposes: generating candidate solutions via ensemble reasoning, inspiring creative approaches, reflecting on previous attempts, adjusting the complexity of approaches, imposing problem constraints, and addressing domain-specific challenges.

**Dynamic Hints**

These hints use an internal LLM to dynamically generate multiple candidate solutions and refine them through debate and synthesis.

- **CandidateSelectionHint:** Generates several candidate solutions for a given problem and then uses a two-step process (first, detailed reasoning for each candidate; second, synthesis of the best solution) to present the top candidate. This hint leverages step-by-step reasoning to help decide among multiple possible approaches.

- **DebateHint:** Uses a debate format where opinions are generated from multiple historical figures (or personas) about a problem. It then synthesizes these divergent views into a concise, best possible solution. This hint is ideal when diverse perspectives might reveal hidden insights into the solution.

### Inspiration Hints

These hints are designed to spark creativity by encouraging the solver to leverage domain-specific expertise or past successful strategies, including a prompt for getting inspired by prior approaches.

- **AlgebraHint:** Invokes algebraic techniques and principles, helping the solver to explore a variety of functions and relationships.

- **GameTheoryHint:** Draws on strategic decision-making principles from game theory, offering insights into competitive or adversarial problem settings.

- **RLRewardFunctionsHint:** Utilizes ideas from reinforcement learning, specifically around optimizing reward functions, to enhance solution approaches.

### Reflection Hints

These hints encourage self-assessment and iterative improvement by prompting the solver to reflect on both successes and mistakes from prior attempts.

- **ReflectAndAvoidErrorsHint:** Advises reflecting on previous mistakes and learning from them to prevent similar errors in future attempts.

- **IdentifySuccessesHint:** Encourages the solver to pinpoint what worked well in earlier attempts and to replicate those successful strategies.

- **CombineIdeasHint:** Suggests merging two or more ideas to create a novel approach that benefits from multiple insights.

- **SeekDeeperInsightsHint:** Prompts the solver to look beyond the obvious and uncover hidden connections or deeper insights in the problem.

### Complexity Hints

These hints help modulate the difficulty of the approach, suggesting strategies to simplify or to challenge the solver with more rigorous methods.

- **TryEasyHint:** Suggests trying a simpler or more straightforward approach.

- **TryEasierHint:** Recommends opting for an even simpler variant than before, reducing complexity further.

- **TryHardHint:** Encourages the solver to explore a challenging strategy that might lead to more robust solutions.

- **TryHarderHint:** Urges the solver to ramp up the challenge, trying an approach more difficult than previous attempts.

**Diversity Hints**

These descriptions are designed to provide clear guidance on how each hint supports diverse thinking and problem-solving techniques.

- **TryDifferentHint:** Advises experimenting with a markedly different strategy compared to those used before, potentially uncovering a new pathway.

- **AlternativePerspectiveHint:** Invites the solver to rethink the problem from a different angle, potentially revealing non-obvious solutions.

**Limiting Hints**

These hints impose specific constraints to ensure the solution remains within manageable or expected bounds.

- **LimitVariablesHint:** Directs the solver to restrict the formula to exactly a given number of variables, ensuring simplicity or focus in the formulation.

**Sparsity Domain Specific Hints**

Aimed primarily at problems involving matrix operations or when matching output dimensions is critical, these hints are tailored specifically to the task at hand.

- **ComplementMatchingSizeHint:** Advises a step-by-step approach: develop a novel formula, evaluate its size against an expected matrix size, and only proceed if sizes match—otherwise, adjust operations accordingly.

- **MatrixMultiplicationHint:** Recommends using matrix multiplication by listing potential components with their respective output shapes, ensuring that the final result meets the expected dimensions.

- **NormalizationHint:** Suggests incorporating normalization techniques (e.g., Min-Max Scaling, Z-Score, L2 Norm, L1 Norm) to refine the solution.

- **ResultDimensionHint:** Ensures that the final formula outputs a matrix or result with the precise dimensions required by the problem.

- **UnaryOperationsHint:** Proposes using one or more unary operations (such as squaring, negation, absolute value, logarithm, exponential, etc.) to adjust the result, emphasizing the importance of adapting operations to meet the problem's dimensional needs.

# I    Implementation details

**Prompt template for formula generation**

At every generation step, we combine the Prompt Template with the Instruction Message, and optionally include a Hint Message.

**Prompt Template**

```
1   <ROLE_PROMPT>
2
3   <PROBLEM DESCRIPTION>.
4
5   When you respond, output a JSON where the first key ("thought") corresponds to your thought process
    ↪   when designing the next function.
```

```
6   The second key ("name") corresponds to the name of your next function.
7   The last key ("code") corresponds to the exact python code that you would like to try.
8
9   Here is an example:
10  '''
11  <EXAMPLE JSON FORMAT>
12  '''
```

### Role Prompt

```
1   You are expert in datascience and machine learning. You are highly skilled in creating of new math
    ↪   formulas.
```

### Problem Description

```
1   Your task is to create new formula for LLM unstructured sparsity importance matrix.
2
3   You can use following matrixes to form new formulas, each matrix is associated with its size:
4   ```
5   <AVAILABLE_OPERATIONS>
6   ```
```

### Available operations

```
1   ### Weights
2   - W (512, 1024)
3       Weights of the model.
4
5   ### Activations
6
7   - A.mean (1024, 512)
8     Mean of the activations across the batches.
9
10  - A.M2 (1024, 512)
11    Accumulated sum of squared differences from the mean (used for calculating variance).
12
13  - A.sum_squares (512, 1)
14    Sum of squares of the activations along the first dimension (batch dimension).
15
16  - A.sum_abs (512, 1)
17    Sum of absolute values of the activations along the first dimension.
18
19  - A.min (1024, 512)
20    Minimum value of the activations across the batches.
21
22  - A.max (1024, 512)
23    Maximum value of the activations across the batches.
24
25  - A.mean_abs (512, 1)
26    Mean of the absolute values of the activations along the first dimension.
27
28  - A.mean_squared (512, 1)
29    Mean of the squared values of the activations along the first dimension.
```

```
30
31   - A.variance (1024, 512)
32     Variance of the activations, computed from M2.
33
34   - A.std (1024, 512)
35     Standard deviation of the activations, computed as the square root of the variance.
36
37   ### Gradients
38
39   - G.mean (512, 1024)
40     Mean of the gradients across the batches.
41
42   - G.M2 (512, 1024)
43     Accumulated sum of squared differences from the mean (used for calculating variance).
44
45   - G.sum_gradients (1024, 1)
46     Sum of gradients along the first dimension (batch dimension).
47
48   - G.sum_abs_gradients (1024, 1)
49     Sum of absolute values of the gradients along the first dimension.
50
51   - G.sum_gradients_squared (1024, 1)
52     Sum of squares of the gradients along the first dimension.
53
54   - G.mean_gradients (1024, 1)
55     Mean of the gradients along the first dimension.
56
57   - G.mean_abs_gradients (1024, 1)
58     Mean of the absolute values of the gradients along the first dimension.
59
60   - G.mean_gradients_squared (1024, 1)
61     Mean of the squared gradients along the first dimension.
62
63   - G.variance (512, 1024)
64     Variance of the gradients, computed from M2.
65
66   - G.std (512, 1024)
67     Standard deviation of the gradients, computed as the square root of the variance.
68   '''
```

**Instruction Message**

```
1   Write a function that takes weights, activations and gradients as input and returns a new
    ↪   importance matrix. Each of matrix is Numpy array. Make sure output shape from your function is
    ↪   equal to expected matrix shape <EXPECTED_SIZE>."
```

**Hint Message**

```
1   For this generation make sure you follow these steps:
2   '''
3   <HINT>
4   '''
```

**Formula format**

A sparsity formula is defined as a function that consumes three classes of signals: statistics of weights, statistics of activations, and statistics of gradients. The complete enumeration of available statistics is provided in Table E. As a concrete example, we illustrate a simple magnitude pruning implementation.

```python
def importance_matrix(W, A, G):
    import numpy as np
    return np.abs(W)
```

**Debugging**

Debugging starts with the LLM annotating the code with matrix sizes to enable better troubleshooting. Then the LLM fixes the code. Because the code must be self-contained, we add an additional step that inlines all required imports directly inside the method body.

**Formula Annotation**

```
Your goal is to evaluate operations step by step.
Here are sizes of following matrices:
<AVAILABLE_OPERATIONS>
Think step by step.
Make sure to evaluate each operation one by one printing all input and output
matrix shapes, to find where is the error.
Then use this with simplest way possible to fix issue.
```

**Fixing Formula**

```
Your goal is to evaluate operations step by step.
Here are sizes of following matrices:

'''
<AVAILABLE OPERATIONS>
'''

Think step by step.
Make sure to evaluate each operation one by one printing all input and output
matrix shapes, to find where is the error.
Then use this with simplest way possible to fix issue.

Previous generation failed with following exception

Here is generated method:
'''

'''

Here is exception:
'''
<EXCEPTION>
'''

Here is insightful analysis of matrix sizes which will help you to find the problem:
```

```
26    '''
27    <FORMULA_ANNOTATION>
28    '''
29
30    Use all this information to fix the code
```

**Fixing Imports**

```
1     You are a Python code assistant specialized in correcting import statements
2     in code snippets.
3
4     Please fix the provided Python code by moving all import statements inside
5     of function.
6     The task is fix the import statements and nothing else.
7     Example:
8     import ...
9     def sort():
10        ...
11
12    Correct:
13
14    def sort():
15        import ...
16        ...
17
18    Here is the code:
19    '''
20    
21    '''
```

## J  Licenses

The datasets and tools used in this research are licensed as follows: WikiText is licensed under the Creative Commons Attribution-ShareAlike 3.0 (CC BY-SA 3.0) License, allowing free use, modification, and distribution, with the requirement for attribution and the condition that derivatives must be shared under the same license. SmolLM2 is licensed under the Apache License 2.0, permitting free use, modification, and distribution, including for commercial purposes, provided that attribution is given, a notice of changes is included, and there is no warranty. C4 is licensed under the Open Data Commons Attribution License (ODC-BY), allowing for free use, modification, and distribution, with the condition that attribution is provided and compliance with the terms of the original Common Crawl dataset is ensured. Additionally, EleutherAI's LM Evaluation Harness is open-source software released under the MIT License, which permits free use, modification, and distribution, including for commercial purposes, with attribution required and no warranty.

