# OpenReview forum: "LLM-Guider: A Language-Guided Discovery of Symbolic Pruning Metrics for Post-Training Sparsity in LLMs"
_TMLR — Accepted by TMLR_

### Review · Reviewer_fa1X · 2026-04-09

**Summary Of Contributions:**

The paper introduces LLM-Guider, a framework that adopts LLMs to discover symbolic pruning metrics. The authors define a comprehensive search space, and an LLM explores this space in an iterative loop. Through this process, LLM-Guider discovers for a 135M LLM novel pruning formulas that outperform training-free unstructured pruning baselines.

**Audience:**

Yes

**Audience Explanation:**

The pruning metrics seem novel and valuable to researchers working on sparsity. Further, the methodology of employing LLMs to automate heuristic discovery is also a valuable contribution to the research community.

**Claims And Evidence:**

No

**Claims Explanation:**

The claims are partially supported. Most of the methodology makes sense, but there are some drawbacks in the evaluations that make it difficult to appreciate the concreteness of the claims:
- Model scale: The empirical evidence is limited to models at the ~135M parameter count scale. Most relevant literature includes evaluations on models with 7B parameters and beyond. Without evaluations on larger models, there is no evidence that the discovered formulas maintain their gains.
- Sparsity ratios: While NVIDIA's N:M sparsity corresponds to 50% sparsity, the SparseGPT included evaluations across different sparsity levels, whereas LLM-Guider only evaluated 50% sparsity level. Evaluating on more sparsity levels would ensure that the metric is robust across different compression schemes.

Note: I am not an expert on sparsity, so the authors should push back if anything in my review is inaccurate.

**Requested Changes:**

- [Critical] Evaluations on larger models from other model families: The evaluations are constrained to only a single model with 135M parameters. (Nit: It seems that Appendix B only has a single title.) The paper needs at least one standard-sized LLM (e.g., something that's 7-8B in size) to verify that the formulas scale.
- [Critical] Evaluations on different sparsity levels. Right now, it seems that only 50% sparsity level is evaluated. If other sparsity levels are difficult to evaluate, the authors should explicitly state a justification.
- The appendix includes a bunch of experiments that are quite insightful and central to understanding the evaluations. I would encourage the author to add forward pointers in the main text that reference these sections.
- Grammar nits, including but not limited to:
	+ Intro: " magnitude(Cheng"
	+ Table 3 caption: "SmoLM2-135M"

---

### Review · Reviewer_Kiuj · 2026-04-13

**Summary Of Contributions:**

Authors of this paper proposed a generic language-guided symbolic formula optimization framework called LLM-Guider for post-training unstructured sparsity in LLMs. It operates through three distinct stages, leverages LLM to efficiently explore the formula space, and can tailor for discovering optimal pruning criteria. Extensive analyses were conducted and showed that the formulas discovered by LLM-guider outperform baselines.

**Audience:**

Yes

**Audience Explanation:**

The proposed approach for post-training unstructured sparsity in LLMs can be practical in resource-constrained settings. The proposed framework by iterating the generation and selection of formulas using LLMs can be interested to broader audience.

**Broader Impact Concerns:**

There is no concern.

**Claims And Evidence:**

Yes

**Claims Explanation:**

LLM is used to generate and selection symbolic formulas for post-training unstructured sparsity in LLMs. The iterative process is expected to optimize the symbolic formula for target problem based on the referred data. The proposed framework discovered two effective pruning formulas through extensive experimentation on SmolLM model. Experimental results on language modeling and zero-short evaluation shows the better performance than baselines. However, the proposed iterating process is a bit ad hoc since the investigation of termination criterion and convergence analysis is missing.

**Requested Changes:**

Some details in Figure 1 comparing to the content in section 3 could be further clarified such as the formula and the float value (score?), the k short formula selection, the concept of node, etc.

The three distinct stages operated by LLM-guider are iterated through some predefined number of iterations. In evaluation setup, 100 generation evaluation per run was used. It is interesting to see if the iterating process can consistently improve the effectiveness of the model, and how to determine if the performance of the model is convergent or divergent.

The proposed framework was evaluated on SmolLM2. It is interesting to see if it works for other models, and the discovered formulas on different models could be consistent or not In other words, the generalization among models could be investigated further.

---

### Review · Reviewer_dbWE · 2026-04-21

**Summary Of Contributions:**

The paper introduces LLM-Guider, a language-guided symbolic optimization framework that discovers pruning metrics for post-training unstructured sparsity in LLMs. Instead of relying on manually designed heuristics or purely evolutionary search, the method uses an LLM-driven loop consisting of k-shot example selection, prompt-based formula generation, and iterative evaluation to explore the space of symbolic importance metrics. A key idea is to leverage language prompts (including hints and in-context examples) to inject domain knowledge into the search, while maintaining interpretability through explicit formulas. Empirically, the discovered formulas outperform standard baselines such as magnitude pruning, Wanda, and PrunerZero, and approach SparseGPT without requiring weight updates.

Strengths include a clean and modular framework, interpretable outputs (symbolic formulas), and a convincing demonstration that LLM-guided search can go beyond fixed search spaces. Weaknesses include limited evaluation scope (single main model family and sparsity level), potential overfitting of discovered formulas to the calibration setup, and unclear novelty relative to prior LLM-guided search works beyond the specific application domain.

**Audience:**

Yes

**Audience Explanation:**

Yes. The work sits at the intersection of model compression, automated discovery, and LLM-based reasoning, which are all active areas of interest in the community. In particular, the idea of using LLMs as a structured search mechanism for discovering interpretable formulas is broadly relevant beyond pruning, aligning with recent trends in LLM-guided scientific discovery and optimization. Researchers working on sparsity, efficient inference, or neural architecture/search methods would find the approach useful, especially due to its transparency compared to black-box search methods. That said, the appeal may be somewhat niche, as the contribution is incremental within the growing body of LLM-guided search frameworks, and its impact depends on whether the approach scales to larger models and more realistic deployment settings.

**Claims And Evidence:**

Yes

**Claims Explanation:**

The main empirical claim that LLM-Guider discovers pruning metrics that outperform existing heuristic and search-based methods is reasonably supported within the presented experimental setup. The paper provides comparisons on WikiText-2 perplexity and zero-shot benchmarks, where the proposed method consistently improves over Magnitude, Wanda, and PrunerZero, and is competitive with SparseGPT.

However, the evidence is somewhat narrow: experiments are primarily conducted on a single small model (SmolLM2-135M) at a fixed sparsity ratio, and while there is a brief cross-model evaluation, it remains limited in scale. Additionally, improvements over strong baselines are modest in some settings (e.g., close to SparseGPT), and statistical significance or robustness across seeds and datasets is not deeply analyzed. Overall, the evidence is directionally convincing but not fully comprehensive for broad claims about general effectiveness.

**Requested Changes:**

- Broaden empirical validation:evaluate on multiple model scales (e.g., >1B parameters), different sparsity ratios, and additional datasets to demonstrate robustness and generality of the discovered formulas.
- Strengthen comparison to prior LLM-guided discovery methods (e.g., Eureka, symbolic optimizer discovery) by clarifying what is fundamentally new beyond applying the paradigm to pruning.
- Analyze generalization vs. overfitting: explicitly test whether discovered formulas transfer without re-search across models and calibration sets, and quantify performance drop if any.
- Provide variance across multiple independent runs of the full pipeline (not just ablations) to assess stability of the discovered solutions.
- Include computational cost analysis (e.g., total GPU time vs. baselines like PrunerZero or SparseGPT) to contextualize practical trade-offs.

---

### Decision · Action_Editor_mW4t · 2026-05-24

**Recommendation:** Accept with minor revision

**Additional Comments:**

The authors did not respond during the discussion phase, and several reviewer concerns therefore remain unresolved. The paper would benefit from a stronger discussion of generalization and robustness, particularly regarding transfer across larger model scales, different sparsity levels, and calibration settings. The final version should also more clearly position the contribution relative to prior LLM-guided symbolic discovery frameworks and better contextualize the computational trade-offs of the proposed search procedure.

**Audience:**

Yes

**Audience Explanation:**

The paper addresses topics of broad current interest, including model compression, LLM-guided discovery, and automated heuristic design, and should interest readers working in efficient LLMs and ML-driven optimization.

**Claims And Evidence:**

Yes

**Claims Explanation:**

The paper provides convincing evidence that the proposed framework can discover effective symbolic pruning formulas that outperform several standard pruning baselines within the evaluated setting. While the empirical scope remains somewhat limited in terms of model scale, sparsity levels, and robustness analysis, the claims are appropriately supported for the demonstrated regime.